# Impact of COVID-19 on Urology Practice: A Global Perspective and Snapshot Analysis

**DOI:** 10.3390/jcm9061730

**Published:** 2020-06-03

**Authors:** Stavros Gravas, Damien Bolton, Reynaldo Gomez, Laurence Klotz, Sanjay Kulkarni, Simon Tanguay, Jean de la Rosette

**Affiliations:** 1Department of Urology, Faculty of Medicine, School of Health Sciences, University of Thessaly, 41110 Larissa, Greece; 2Department of Urology, Austin Health, Heidelberg, Victoria 3084, Australia; damienmbolton@gmail.com; 3Universidad Andres Bello, Hospital del Trabajador de Santiago, Santiago 7550196, Chile; gomez.reynaldo@gmail.com; 4Division of Urology, Sunnybrook Health Sciences Centre, Toronto, ON M4N 3M5, Canada; laurence.klotz@sunnybrook.ca; 5Kulkarni Reconstructive Urology Center, Pune 411038, India; sanjaybkulkarni@gmail.com; 6Department of Surgery, Division of Urology, McGill University, Montreal, QC H3A 1A1, Canada; simon.tanguay@mcgill.ca; 7Department of Urology, Istanbul Medipol University, Istanbul 34214, Turkey; j.j.delarosette@gmail.com

**Keywords:** coronavirus, covid-19, pandemic, sars cov-2, urology, practice management

## Abstract

The global impact of the 2019 novel coronavirus disease (COVID-19) pandemic on urology practice remains unknown. Self-selected urologists worldwide completed an online survey by the Société Internationale d’Urologie (SIU). A total of 2494 urologists from 76 countries responded, including 1161 (46.6%) urologists in an academic setting, 719 (28.8%) in a private practice, and 614 (24.6%) in the public sector. The largest proportion (1074 (43.1%)) were from Europe, with the remainder from East/Southeast Asia (441 (17.7%)), West/Southwest Asia (386 (15.5%)), Africa (209 (8.4%)), South America (198 (7.9%)), and North America (186 (7.5%)). An analysis of differences in responses was carried out by region and practice setting. The results reveal significant restrictions in outpatient consultation and non-emergency surgery, with nonspecific efforts towards additional precautions for preventing the spread of COVID-19 during emergency surgery. These restrictions were less notable in East/Southeast Asia. Urologists often bear the decision-making responsibility regarding access to elective surgery (40.3%). Restriction of both outpatient clinics and non-emergency surgery is considerable worldwide but is lower in East/Southeast Asia. Measures to control the spread of COVID-19 during emergency surgery are common but not specific. The pandemic has had a profound impact on urology practice. There is an urgent need to provide improved guidance for this and future pandemics.

## 1. Introduction

Since the first reports of pneumonia of unknown etiology originating in Wuhan City, China reached the World Health Organization (WHO) China Country Office on December 31, 2019, healthcare systems worldwide have rapidly adapted their practices to accommodate what would be officially declared a pandemic on March 11, 2020 [1]. This pandemic, caused by a novel coronavirus SARS CoV-2, is associated with the disease now known as COVID-19.

It rapidly spread from East to West, with the first deaths reported in Asia between 31 December 2019 and 20 January 2020 [1], and then globally to other countries. As of May 25, 2020, a total of 5,439,559 cases of COVID-19 and 345,589 deaths have been confirmed across 188 countries/territories [2,3].

The impact of the pandemic on all medical practitioners is unprecedented, with clinicians forced to postpone elective and low-risk procedures and prioritize high-risk procedures, conduct appointments and patient counseling via virtual platforms, and re-allocate time and efforts to aid emergency departments and intensive care units (ICUs). To reduce human-to-human contact and resultant viral transmission between patients and healthcare workers, patient cases have been evaluated and triaged based upon risk level and prognosis. 

The Société Internationale d’Urologie (SIU) is a major global society for sustainable urological education and collaborative philanthropic activities aimed at improving urological care. Its mission is to enable urologists in all nations, through international cooperation in education and research, to apply the highest standards of urological care to their patients. The SIU website is https://www.siu-urology.org/.

This report by the SIU aims to detail the impact of COVID-19 on global urology practices, with an effort made to explore differences in trends across regions and practice settings.

## 2. Experimental Section

The SIU was founded in 1907 and currently has 10,018 members in 131 countries. Members of the SIU Executive Board comprised of 27 members from all continents designed a survey titled “Urology in the time of COVID-19”. This survey comprised multiple-choice questions about respondent demographics and practice change in response to COVID-19, the answers to which are analyzed and presented here. It also included questions about specific procedures, educational needs, and concerns about contracting COVID-19 that, because they represent different domains, are not included in the present work and will instead be addressed in a separate future analysis. The full survey is available in Appendix A. 

The survey was opened on March 27, 2020, and closed on April 11, 2020. It was administered online using the Aventri™ platform (Connecticut, USA). Distribution of the survey took place via email, using names on the SIU eNews mailing distribution list. It was posted on the SIU blog, available at https://www.siu-urology.org/siu-news/archives (AIKI CMS for SIU, Montréal, Canada) as well as on the SIU Academy website, https://academy.siu-urology.org (Multilearning, Montréal, Canada). The survey included reasons why it was being conducted and the importance of participation. No compensation was offered for its completion. All responses were anonymous.

In order to facilitate analysis of the impact of COVID-19 on healthcare settings as it spread from East to West, respondents were grouped into the following regions: East/Southeast Asia and nearby regions, West/Southwest Asia and nearby regions, Europe, Africa, North America, and South America. The list of countries included in each region is provided in Appendix A.

Regional variations in responses were explored in order to capture evolving changes in the impact of the pandemic on practice as it progressed across the globe from East to West. Variations in response based on practice setting were also evaluated for some questions. 

### Statistical Analysis 

Descriptive statistical analyses (absolute number and percentages) were conducted on the final survey results to determine background demographics and the proportion of respondents who answered each option in the multiple-choice questions. 

The final survey data were explored via a series of omnibus Pearson chi-square tests on responses to the multiple-choice questions, each crossed with the geographic region or practice setting factors (using a standard alpha threshold of *p* = 0.05). Statistically significant differences thought to be of practical interest were further explored by calculating, for each cell in a contingency table, the adjusted standardized residuals. Conceptually, these are the Z-transformed differences between the expected and observed percentage for that cell (see Agresti, 2013 [4]). By examining the adjusted standardized residuals, we determined which observed cells showed a higher/lower percentage than expected under the null hypothesis (no statistically significant relationship between the geographical region or practice settings factors and responses to the multiple-choice questions). The statistical significance threshold for each contingency table was Bonferroni corrected for the number of tests conducted within that table (this is standard practice to reduce the risk of Type I error with multiple comparisons).

## 3. Results

### 3.1. Background Demographics

In total, 2494 self-selected, non-representative, non-probability participants completed the survey. Table 1 lists their background demographics. Overall, 798 respondents (32.0%) were aged 39 or younger, 1057 (42.4%) were 40–55 years old, and 639 (25.6%) were 55 or older. The respondents originated from 76 countries. Nearly half of the respondents (1161 [46.6%]) worked in an academic setting, with 719 (28.8%) in private practice and 614 (24.6%) in the public sector.

### 3.2. Impact on Practice

#### 3.2.1. Outpatient Consultations

Figure 1 demonstrates regional trends in changes to outpatient consultations in the clinic setting based on whether they are fully operational, completely shut down, or modified (i.e., telephone consultations only, in-person follow-ups only, or a combination of the two). The modified consultations approach was used to avoid false positive results from the numerous comparisons of the categorical response options.

Chi-square analysis revealed that consultations at the outpatient clinic or office varied by geographical region (X210 = 792.22, *P* < 0.001). A follow-up examination of the adjusted standardized residuals, using a Bonferroni corrected threshold of Z = ± 2.99 (corresponding to *P* = 0.003), revealed that reports of completely locking down the outpatient clinic or office were lower than expected in East/Southeast Asia (1.6%; adjusted standardized residual = −7.78) and higher than expected in South America and West/Southwest Asia (20.2%, adjusted standardized residual = 3.23, and 21.0%, adjusted standardized residual = 5.21, respectively). 

Reported use of a modified approach to the management of the outpatient clinic (i.e., telephone consultations only, in-person follow-ups only, or a combination of the two) was lower than expected in East/Southeast Asia (40.4%, adjusted standardized residual = −15.91) and higher than expected in Europe (82.7%, adjusted standardized residual = 10.83). 

Fully operational consultations (i.e., no change from pre-COVID-19 times) were higher than expected in East/Southeast Asia (58.05%, adjusted standardized residual = 26.87) and lower than expected in West/Southwest Asia (7.3%, adjusted standardized residual = −4.99), Europe (4.7%, adjusted standardized residual = −13.23), and South America (2.0%, adjusted standardized residual = −5.53).

#### 3.2.2. Emergency Surgical Procedures

Emergency cases requiring immediate surgery were given increased attention but without specific measures in place, according to 40.9% of respondents. One third (33.2%) reported that patients are managed as if they were COVID-19 positive, 10.7% reported no change in practice, and 8.2% test patients for COVID-19 and then proceeded without special precautions (Figure 2). 

When comparing regions, the greatest proportion sent for COVID-19 testing and then proceeded without special precautions in Europe (10.1%) as well as in the East/Southeast and West/Southwest Asia regions (9.3% for each). The regions most likely to report no change in their approach were East/Southeast Asia (15.6%) and Africa (14.4%). Proceeding as if the patient was infected with COVID-19 was most commonly reported in North and South America (43.7% and 48.5%, respectively) and least commonly reported in East/Southeast Asia and Africa (26.5% and 22.5%, respectively) (Figure 2).

#### 3.2.3. Elective Surgical Procedures

As seen in Figure 3, access to the operating room (OR) for elective surgery was reported to be completely cut off (37.6%) or reduced by >75% (30.2%) by the majority of respondents. 

Chi-square analyses revealed differences in access to the OR by geographical region (X225 = 711.93, *P* < 0.001) and practice setting (X210 = 33.79, *P* < 0.001).

A follow-up examination of the adjusted standardized residuals for differences by geographical region, using a Bonferroni corrected threshold of Z = ± 3.20 (corresponding to *P* = 0.001), revealed higher-than-expected reports of no change in OR access in East/Southeast Asia (26.4%; adjusted standardized residual = 16.84). The rates of reported no change were lower than expected in West/Southwest Asia (2.6%; adjusted standardized residual = −3.89), Europe, (3.1%; adjusted standardized residual = −7.10), and South America (0.5%; adjusted standardized residual −3.84).

Examining differences by practice setting using a Bonferroni corrected threshold of Z = ± 2.99 (corresponding to *P* = 0.03) revealed lower than expected reports of <25% reduction in the private setting (3.6%, adjusted standardized residual = −3.59).

The responsibility for deciding which patient should be operated on in the COVID-19 era fell to the urologist according to 40.3% of respondents, the department chair according to 21.5%, a committee of the surgical division according to 19.2%, or the chair of the hospital board according to 11.4% (Figure 4).

The chi-square analysis revealed regional differences (X220 = 228.33, *P* < 0.001) as well as differences by practice setting, (X28 = 201.95, *P* < 0.001) with respect to who made decisions about OR access.

Using a Bonferroni adjusted threshold of Z = 3.14 (corresponding to *P* = 0.002), urologists themselves were reported to be the decision makers at higher-than-expected rates in East/Southeast Asia (57.4%, adjusted standardized residual = 8.06) and West/Southwest Asia (50.0%, adjusted standardized residual = 4.23) and at lower than expected rates in Europe (28.3%, adjusted standardized residual = −10.62). A committee of the surgical division had decision-making responsibility at higher-than-expected rates in Europe and North America (25.6%, adjusted standardized residual = 7.01; 30.1%, adjusted standardized residual = 3.91, respectively). The department chair was reported to have the decision-making power at higher-than-expected rates in Europe (27.9%, adjusted standardized residual = 6.81).

Explorations by practice setting using a Bonferroni adjusted threshold of Z = 2.94 (corresponding to *P* = 0.003) revealed greater than expected reports that urologists themselves have the decision-making power in the private setting (50.6%, adjusted standardized residual = 6.69). The department chair was reported to be the decision maker at higher-than-expected rates in the academic setting (29.7%, adjusted standardized residual = 9.33 and 5.7%).

Elective operations that may require transfusion were performed only if patients were at high risk of disease progression, according to 47.3% of respondents. The remainder reported that these procedures were postponed (30.7%), performed as in the past (16%), or replaced with minimally invasive procedures when possible (6.0%). Similar patterns were seen for surgeries that might have required admission to the ICU, with equivalent responses by 43.7%, 41.5%, 9.9%, and 4.9%, respectively.

## 4. Discussion

It was hypothesized that differences in how urology practice is influenced by the COVID-19 pandemic, as well as efforts to adapt practice, are likely influenced by regional variation in the severity of the COVID-19 situation, including how healthcare systems are organized globally and what resources are available. This survey and analysis by the SIU were designed to provide a snapshot of how the COVID-19 pandemic has impacted urology practices as it spread globally from East to West with respect to outpatient consultation, emergency surgical procedures, and elective surgical procedures.

Our findings illustrate substantial differences in health policies, which match the spread of COVID-19 from East to West. In East/Southeast Asia, healthcare systems appear to be the closest to returning to normal (China announced the end to COVID-19 lockdown in Wuhan on April 7, 2020), with the greatest proportion of reports that outpatient consultations were operating without restriction coming from this region. 

A high proportion of reports of a modified approach to outpatient consultations, including telephone consultations, restricting in-person visits to follow-ups only, or a combination of the two, may reflect the pressure placed on healthcare systems during the pandemic to develop telemedicine platforms. This modified approach was common in all regions except East/Southeast Asia and was most frequently reported in Europe. 

The need to conduct emergency surgery is common in the urology setting. It remains unclear how best to proceed in such circumstances. 

Uncertainty in this area was reflected in the survey findings, with over one-third of respondents reporting that they give emergency cases increased attention but without following specific measures. Still, as many as one-third proceeded as if patients were infected, which was most commonly reported in North and South America, and least often reported in Africa, with the latter possibly reflecting a lack of resources.

Only a small proportion of respondents reported that patients undergoing emergency surgery were tested for COVID-19. This practice was reported most often in Europe as well as the two Asian regions.

As with outpatient consultations, reports of no change in the approach to emergency surgeries were highest in East/Southeast Asia again reflecting a more rapid return to normal operations in that region.

A similar trend is witnessed in the case of elective surgery. The majority of respondents reported that access to the OR for elective surgery was completely cut off or restricted by >50%. The driving forces behind the practice may include the need for healthcare systems to conserve resources in terms of personnel and available ventilators and the need to decrease the risk of patients being exposed to SARS-CoV-2 in the hospital environment. Once again, an unmodified, unrestricted approach was reported most often in East/Southeast Asia. Lower than expected reports of <25% reduction in access to the OR were reported in the private setting, compared with academic or public non-academic/military settings.

Regional and practice setting variations with respect to who bears the responsibility for deciding which patient should be operated on in the COVID-19 era may reflect differences in organizational structures and their effects on coordinating compliance with expert recommendations and implementation of guidelines. Overall, urologists themselves were most often reported to bear this responsibility, and this was reported at higher-than-expected rates in Eastern and Western Asia regions as well as in the private setting. 

Surgical division committees were reported to have this decision-making power at higher-than-expected rates in Europe and North America. The department chair had the decision-making power, according to respondents, more often in Europe as well as in the academic setting. Elective surgeries that may require transfusion or ICU care were typically postponed or performed only if patients are at high risk of disease progression. Such decisions may be influenced by local blood bank systems, the shortage of blood products due to limited blood donations in the COVID-19 era, and the availability of ICU wards. 

This survey has several limitations. Many invited urologists may not have taken part in the survey due to the number of other surveys exploring practice change in the face of the COVID-19 pandemic. Moreover, the participation from some continents may be limited by language and access barriers. 

Furthermore, the survey was conducted in many countries, which are at different phases of the pandemic, with differences in terms of speed of action of adopted measures and strategies. This may explain the smaller number of responses from Africa and the Americas since the disease spread later in these regions. In addition, even in the same country, there might be regional differences in the spread of COVID-19.

The strength of the present survey is its global nature and the fact that it presents a perspective from specialists involved in both surgical and clinical activities. Moreover, this is the largest number of respondents to a surgical survey thus far addressing professional concerns in the COVID-19 era.

## 5. Conclusions

Our survey reveals significant regional and practice setting variations on the impact of the COVID-19 pandemic on urology practice. While healthcare systems in East/Southeast Asia are beginning to return to their pre-pandemic practices, the rest of the world reports a high degree of a lockdown or modified practice. Increased use of telemedicine represents a major shift in the approach to outpatient follow-up. Worldwide, urologists themselves bear a great deal of responsibility when deciding who should undergo surgery in the non-emergency setting. This is particularly true for urologists working in private settings. The development of evidence-based strategies for mitigating the risk of COVID-19 spread or a similar future pandemic is urgently needed.

## Figures and Tables

**Figure 1 jcm-09-01730-f001:**
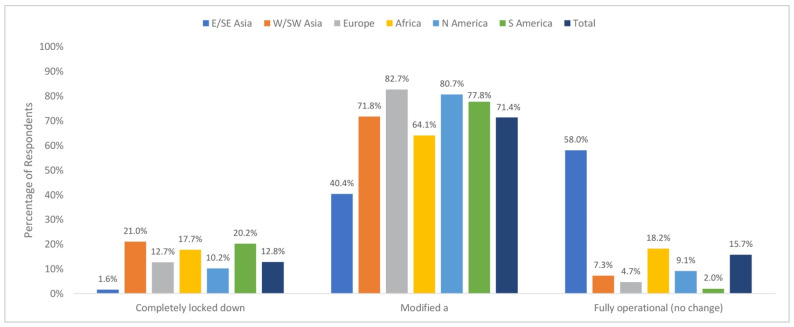
Changes in consultations in the Outpatient Clinic by Region—Completely. Locked vs. Modified vs. Fully Operational. a: Includes Follow-up only, replaced by phone/tele-consult, or combination of both.

**Figure 2 jcm-09-01730-f002:**
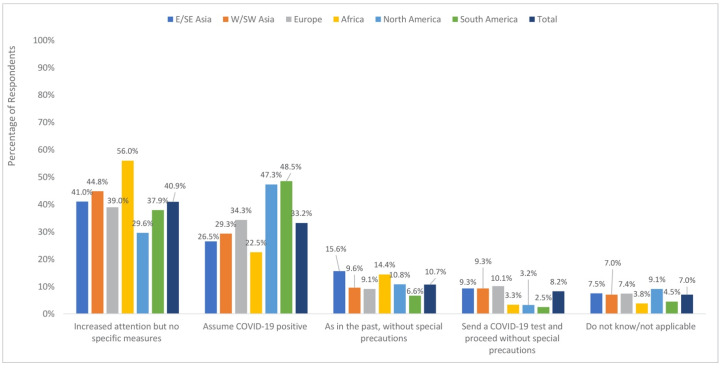
Approach to the management of emergency cases requiring immediate surgery by region.

**Figure 3 jcm-09-01730-f003:**
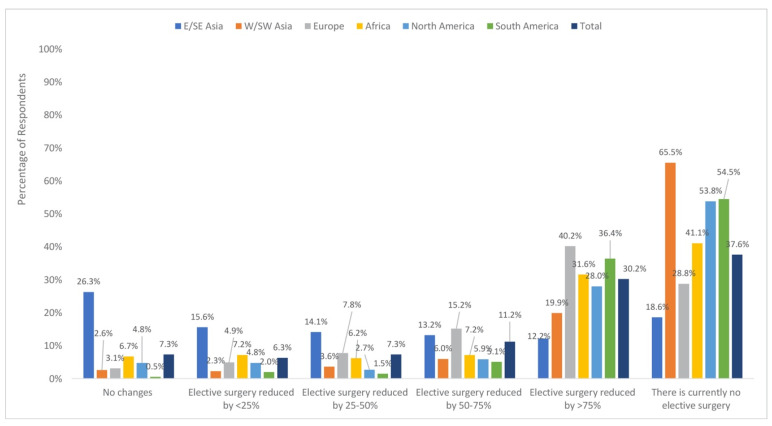
Access to the operating room for elective surgeries by region.

**Figure 4 jcm-09-01730-f004:**
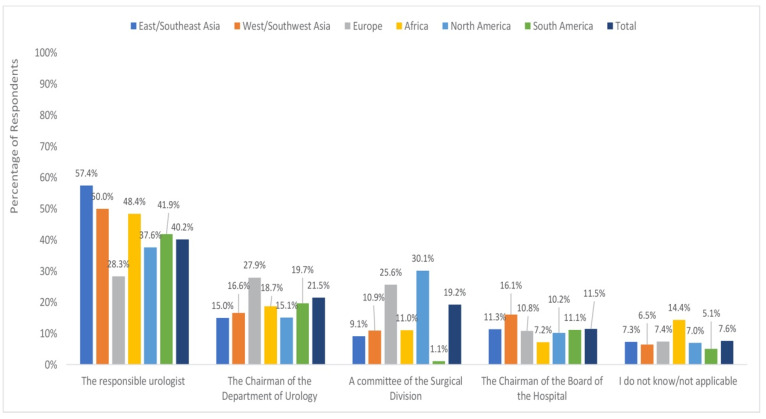
Responsibility for deciding which operations take place by region.

**Table 1 jcm-09-01730-t001:** Background demographics.

Variable	*N* (%)
**Total**	2,494 (100)
**Age (Years)**	
<40	798 (32.0)
40–55	1,057 (42.4)
>55	639 (25.6)
**Region of Origin**	
Europe	1,074 (43.1)
East/Southeast Asia	441 (17.7)
West/Southwest Asia	386 (15.5)
Africa	209 (8.4)
South America	198 (7.9)
North America	186 (7.5)
**Practice Setting**	
Academic/University Hospital	1,161 (46.6)
Private Practice (Office/Hospital)	719 (28.8)
Public Non-Academic/Military/Veterans’ Hospital	614 (24.6)

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
