# Peer review of "Impact of COVID-19 on Urology Practice: A Global Perspective and Snapshot Analysis"

_jcm, 2020, doi:10.3390/jcm9061730_

Round 1

Reviewer 1 Report

The authors describe the impact of the Covid pandemia on urology practice as revealed by an online survey of about 2.500 urologists world wide. Overall, an article very actual in the time being revealing different attitudes according to the different countries.

Introduction:

  • Depending on the publication date the number of cases and deaths due to COVID should be the most actual achievable

Results:

Outpatient consultations

figure 1: modified consultations - what are „in person-follow ups“ ? if this means a visit with personal contact I would suggest to split up the section „modified a“ into „personal contacts“ and „non-personal contacts (telephone, online consultation)“

is there a correlation between the answers of the questionnaire and the severity of the epidemic in the respective country ? Time between first Corona detection and answering the questionnaire may be relevant , as countries may develop a more moderate and liberal handling of patient contacts either at the beginning of the epidemic or far after their climax.

Emergency surgical procedures

  • Again, correlation between infection rate in the country and measures taken in treatment ?
  • Any numbers on correlation between infection of hospital staff and aproach to procedures ?

Reviewer 2 Report

The authors did well on valuable and difficult research.
In the CoVID-19 era, changes in care patterns are essential and there will be policies for each hospital.
There are no major opinions in analysis and results.
However, even in the same country, the incidence of diseases varies by region, but this study does not reflect the characteristics of pandemic area. Please mention this concern in the Discussion part.
